# A Noisy nnU-Net Student for Semi-supervised Abdominal Organ Segmentation

Gregor Koehler[1,2][0000−0002−5263−6786],
Fabian Isensee[1,2,3][0000−0002−3519−5886], and
Klaus Maier-Hein[1,2][0000−0002−6626−2463]

[1] German Cancer Research Center (DKFZ)
[2] Division of Medical Image Computing, German Cancer Research Center,
Heidelberg, Germany
[3] HI Applied Computer Vision Lab
g.koehler@dkfz.de

**Abstract.** While deep learning methods have shown great potential in the context of medical image segmentation, it remains both time-consuming and expensive to collect sufficient data with expert annotations required to train large neural networks effectively. However, large amounts of unlabeled medical image data is available due to the rapid growth of digital healthcare and the increase in availability of imaging devices. This yields great potential for methods which can exploit large unlabeled image datasets to improve sample efficiency on downstream tasks with limited amounts of labeled data. At the same time, deploying such models in real-world scenarios poses some limitations in terms of model size and required compute resources during inference. The 2022 MICCAI FLARE Challenge tries to address both these aspects in a task where participants can make use of 2000 unlabeled, as well as 50 labeled images, while also measuring inference speed, CPU utilization and GPU memory as part of the evaluation metrics. In the context of this challenge, we propose a simple method to make use of unlabeled data: The noisy nnU-Net student. Here the unlabeled data is exploited through self-training, where a teacher model creates pseudo labels, which in turn are used to improve a student model of the same architecture. We show, based on results in a cross-validation and a separate held-out dataset, that this simple method yields improvements over even a strong baseline ($+2$ DSC), while simultaneously reducing inference time by an order of magnitude, from an average of over 500s to roughly 50s, and peak memory requirements by almost a factor of two.

**Keywords:** Organ Segmentation · Self-training · Noisy student

## 1 Introduction

The MICCAI FLARE 2022 challenge is concerned with multi-organ segmentation from abdominal CT scans. The dataset consists of 2000 unlabeled CT scans and 50 labeled CT scans. This represents a challenge often faced in practice,

where medical images are readily available, but expert annotations are scarce due to their expensive and time-consuming nature. A successful method in this context has to make efficient use of the unlabeled images in addition to the labeled set. Additionally, the FLARE 2022 challenge not only evaluates via established segmentation metrics, but also measures CPU utilization and GPU memory during inference. This poses limits on the model size and pre-/post-processing used. Being able to efficiently run inference for segmentation models is often a necessary step for their application in clinical practice, where long inference times can be prohibitive. We address the challenges of this efficiency-focused semi-supervised setting by choosing a moderate architecture size paired with self-training based on noisy student training [9].

The main contributions of this work can be summarized as follows:

– We propose a simple extension to the proven nnU-Net framework using self-training with an ensemble of teacher models and a noisy student model of the same network architecture.
– With some small architecture and preprocessing adaptations, we greatly reduce the memory footprint of the proposed method, while sacrificing only little performance compared with a larger model.
– Through a lightweight inference and resampling scheme, we greatly reduce the resource requirements and inference time compared to nnU-Net.
– We evaluate the effectiveness of the proposed method in the context of the FLARE 2022 challenge, where we achieve performance improvements over even a strong baseline.

## 2   Method

We propose a method based on the nnU-Net framework [5]. To make use of the large unlabeled dataset, we implement noisy student training, inspired by [9], with additional fine-tuning. Additionally, we propose several inference strategies detailed in Section 2.3 to reduce resource consumption during inference.

### 2.1   Preprocessing

Following [5], we make use of the following preprocessing steps:

– Cropping the individual scans to non-zero region.
– Global dataset intensity percentile clipping and z-score normalization with global foreground mean and standard deviation.
– Resampling to median spacing for each axis.

The intensity percentile clipping and normalization based on global foreground mean and standard deviation are employed due to the CT scan values representing physical properties, which should be retained in the preprocessed state.

## 2.2   Proposed Method

The proposed **noisy nnU-Net student** is inspired by noisy student training [4,9], based on the nnU-Net framework [5]. Here, we distill the knowledge of an ensemble of teacher models into the student by using pseudo labels created by the teacher ensemble. In knowledge distillation, the student model is often chosen to be of smaller capacity to aid in scenarios where the teacher model is too expensive to deploy in a real-world use-case. We instead opt to use the same architecture for both teacher and student. This was shown to yield performance improvements in the context of image classification [9]. A schematic overview of the U-Net architecture used is illustrated in Figure 1. This architecture represents a slight adaptation from the full resolution 3D nnU-Net [5,7], where the base number of convolutional filters is reduced to achieve a smaller memory footprint.

We first train the teacher model in a standard nnU-Net training scheme on the labeled portion of the FLARE 2022 challenge dataset. This results in 5 different teacher models trained in a 5-fold cross-validation. To create robust pseudo labels for the following student training, we ensemble the predictions from all 5 teacher models by averaging the softmax outputs of the individual models before creating the hard pseudo labels. For simplicity reasons, we make use of hard pseudo labels to train the student model, instead of soft labels as proposed in [9]. The student model largely follows the same setup. However we sample the training batch by using both labeled samples, as well as unlabeled samples with pseudo labels, in the same minibatch. We balance the two datasets by using 3 times as many samples from the pseudo labeled set in every minibatch. To introduce additional noise for the student, we make use of a stronger data augmentation scheme for the student training, see Section 3.3. In a final stage, we fine-tune the resulting model using only the labeled training cases.

## 2.3   Inference optimization

Apart from the Dice Similarity Coefficient and the Normalized Surface Dice, the 2022 MICCAI FLARE challenge tracks three additional metrics related to inference resource consumption and speed, which contribute to the overall ranking. These metrics are the area under GPU memory-time curve, the area under CPU utilization-time curve, as well as the running time per sample. On top of that, the challenge requires a hard memory limit of 28 GB.

Originally, nnU-Net was not designed with these resource consumption metrics in mind, and instead aims to make efficient use of a given GPU memory setup, optimizing for a maximum of GPU memory and utilization during training to increase segmentation performance. This also impacts nnU-Net's resource consumption during inference. To adapt nnU-Net for the resource consumption metrics used in this challenge, we propose several inference strategies detailed below.

As mentioned in Section 2.2, we adapt the 3D nnU-Net architecture to use fewer filters per convolutional layer. This achieves a smaller memory footprint during both training and inference.

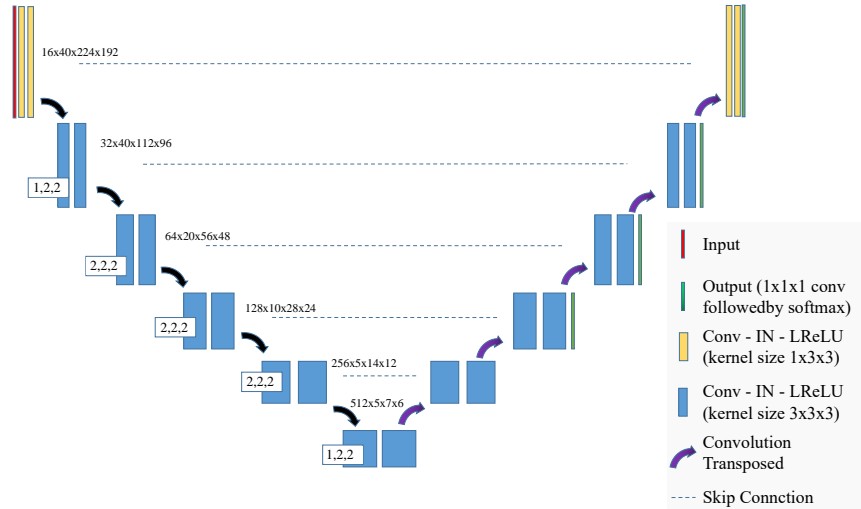

**Fig. 1.** Schematic of the U-Net architecture used for both the teacher and student model. It represents the network architecture proposed by nnU-Net [5], however with half the base number of features. The number of features is then progressively doubled up to the network's bottleneck.

Staying within the challenge's memory limit of 28 GB during inference on at times very large CT scans is a challenging task when using 3D segmentation networks, as the resampling of softmax outputs or segmentation maps to the desired target spacing requires a lot of memory. In this context, we switch the default order of operations usually employed by nnU-Net when creating the final segmentation maps from the network's softmax outputs. Instead of resampling the softmax outputs directly, we instead first create segmentation maps before resampling to the target spacing, as this represents a more computationally efficient operation. Additionaly, we only make use of nearest neighbor interpolation during resampling. While this might come at a segmentation performance cost, this greatly reduces the memory footprint during inference, while simultaneously reducing inference time by roughly an order of magnitude on average. The resulting resource consumption improvements are shown in Table 1. We note that without these adaptations, for 7 out of 50 validation cases, the memory limit would be surpassed. However, these improvements in resource consumption and inference time come at a small cost of increased CPU utilization.

### 2.4   Post-processing

We do not make use of any post-processing in the context of this challenge.

**Table 1.** Comparison of resource consumption metrics during inference on the validation dataset. Best results are highlighted in bold. *Seven cases resulted in peak RAM above 28 GB.

| Inference metric | nnU-Net | low-footprint nnU-Net | noisy nnU-Net student |
|---|---|---|---|
| Mean Peak RAM | 22.69 GB* | 23.07 GB* | **13.78 GB** |
| Mean GPU memory | 5.911 GB | 1.834 GB | **1.722 GB** |
| Mean CPU utilization | **3.87 %** | 3.9 % | 4.38 % |
| Mean inference time | 535.4 s | 122.3 s | **53.5 s** |

## 3   Experiments

### 3.1   Dataset and evaluation measures

As per official challenge documentation, the FLARE2022 dataset is curated from more than 20 medical groups under the license permission, including MSD [8], KiTS [2,3], AbdomenCT-1K [6], and TCIA [1]. The training set includes 50 labeled CT scans with pancreas disease and 2000 unlabeled CT scans with liver, kidney, spleen, or pancreas diseases. The validation set includes 50 CT scans with liver, kidney, spleen, or pancreas diseases. The testing set includes 200 CT scans, where 100 cases show liver, kidney, spleen, or pancreas diseases and the other 100 cases has uterine corpus endometrial, urothelial bladder, stomach, sarcomas, or ovarian diseases. All the CT scans only have image information and the center information is not available.

The evaluation measures consist of two accuracy measures: Dice Similarity Coefficient (DSC) and Normalized Surface Dice (NSD), and three running efficiency measures: running time, area under GPU memory-time curve, and area under CPU utilization-time curve. All measures will be used to compute the ranking. Moreover, the GPU memory consumption has a 2 GB tolerance.

### 3.2   Dataset splits

For model selection, we make use of a 5-fold cross-validation split, resulting in 40 training cases in each fold in the case of the labeled dataset. During the training with the joint labeled and pseudo labeled dataset, we re-use the same cross-validation splits for the labeled samples and add all pseudo labeled scans to each respective fold's training set.

### 3.3   Implementation details

**Environment settings** The environments and requirements are presented in Table 2. We note that while development was done in this environment, the training runs were performed on a GPU cluster node with different hardware.

**Table 2.** Development environment and requirements.

| | |
|---|---|
| Windows/Ubuntu version | Ubuntu 20.04.2 LTS |
| CPU | AMD Ryzen 9 3900X 12-Core CPU@3.80GHz |
| RAM | $4\times16DDR4$@$3.60GHz$ |
| GPU (number and type) | 1 Nvidia GeForce RTX 2080Ti 11G |
| CUDA version | 11.3 |
| Programming language | Python 3.9.7 |
| Deep learning framework | Pytorch (Torch 1.10.2) |
| Specific dependencies | nnunet |

**Training protocols** For training both the teacher and student model, we follow the general training protocol detailed in Table 3. However, the training protocols differ in terms of data augmentation strategies employed. For the initial teacher models, we use the data augmentation strategy described in Table 4. We make use of nnUNet's deep supervision loss based on equally weighted dice and cross-entropy loss terms.

After training the 5 teacher models to convergence, we ensemble their predictions on the unlabeled samples to create pseudo labels for the student model.

**Table 3.** Training protocol.

| | |
|---|---|
| Network initialization | "he" normal initialization |
| Training mode | Mixed precision |
| Batch size | 4 |
| Patch size | $40\times224\times192$ |
| Total epochs | 1000 |
| Optimizer | SGD with nesterov momentum ($\mu = 0.99$) |
| Initial learning rate (lr) | 0.01 |
| Lr decay schedule | Poly (exponent 0.9) |
| Training time (5 initial teacher models) | $5 \times 36$ hours |
| Training time (student model) | 36 hours |
| Training time (fine-tune) | 2.7 hours |
| Loss function | Combined Dice and Cross-Entropy loss |

Using the pseudo labels created by the teacher models, we then train the student model. For this training, we again follow the protocol from Table 3, but use more extensive data augmentation, as detailed in Table 5. We then select the best-performing student model based on a 5-fold cross-validation, and perform an additional fine-tuning using just the labeled training data, again following the protocol from Table 3. The resulting model is used as the final model for test set inference.

**Table 4.** Data augmentation strategy for the teacher model. $p_{sample}$ refers to a probability to apply this augmentation on a sample level, while $p_{channel}$ and $p_{axis}$ are used on a channel and axis level, respectively.

| | |
|---|---|
| Rotation | Angle range: [-30°, 30°] |
| Scaling | Scale range: [0.7, 1.4] |
| Gaussian noise | $p_{sample}$=0.1, $\sigma^2$=[0.0,0.1] |
| Gaussian blur | $p_{sample}$=0.2, $p_{channel}$=0.5, $\sigma$=[0.5,1.0] |
| Brightness (multiplicative) | $p_{sample}$=0.15, Multiplier range: [0.75,1.25] |
| Contrast | $p_{sample}$=0.15, Contrast range: [0.75,1.25] |
| Simulate low resolution | $p_{sample}$=0.25, $p_{channel}$=0.5, Zoom range: [0.5,1.0] |
| Gamma correction | $p_{sample}$=0.1, Gamma range: [0.7,1.5] |
| Mirroring | $p_{sample}$=1.0, $p_{axis}$=0.5 |

**Table 5.** Data augmentation strategy for the student model.

| | |
|---|---|
| Rotation | Angle range: [-30°, 30°] |
| Scaling | Scale range: [0.7, 1.4] |
| Gaussian noise | $p_{sample}$=0.15, $\sigma^2$=[0.0,0.1] |
| Gaussian blur | $p_{sample}$=0.2, $p_{channel}$=0.5, $\sigma$=[0.5,1.5] |
| Brightness (multiplicative) | $p_{sample}$=0.15, Multiplier range: [0.7,1.3] |
| Contrast | $p_{sample}$=0.15, Contrast range: [0.65,1.5] |
| Simulate low resolution | $p_{sample}$=0.25, $p_{channel}$=0.5, Zoom range: [0.5,1.0] |
| Gamma correction | $p_{sample}$=0.15, Gamma range: [0.7,1.5] |
| Mirroring | $p_{sample}$=1.0, $p_{per_axis}$=0.5 |

**Testing protocols** We use the same preprocessing as used during training and use the best student model, as determined in cross-validation, for test set inference. To reduce the memory footprint both for GPU VRAM and RAM, we predict using FP16 precision mode and don't use test-time augmentation, while also up-sampling segmentation maps using nearest neighbor interpolation, as detailed in Section 2.3.

## 4    Results and discussion

### 4.1    Quantitative results for 5-fold cross-validation

We show ablation analysis results of using noisy student training for the low-footprint nnU-Net network in Table 6. In this cross-validation ablation, we can see a noticeable benefit of the proposed noisy student training regarding the mean foreground Dice score. This suggests that self-training with a noisy student can be an effective way to leverage unlabeled images in the context of medical image segmentation.

**Table 6.** Quantitative results for 5-fold cross-validation in terms of mean DSC. The low-footprint nnU-Net refers to the reduced architecture as shown in Figure 1.

| Training scheme | Mean foreground DSC (all classes) |
|---|---|
| low-footprint nnU-Net | $93.5 \pm 0.4$ |
| **low-footprint nnU-Net (w. noisy student training)** | **$94.0 \pm 0.4$** |

### 4.2    Quantitative results on the validation and test set

While cross-validation results yield first performance indicators, we perform a more thorough evaluation of the proposed method on 20 held-out cases. Table 7 illustrates the results on this validation set. As expected, we can observe a clear performance degradation when moving from the original architecture to the low-footprint model, which also represents one individual teacher model used for ensembling the pseudo labels. This is most likely due to the reduced capacity and lack of additional learning signal from the pseudo labeled data. We note that this low-footprint model was necessary to obtain the GPU memory consumption reported in Table 1.

However, the proposed noisy nnU-Net student is able to compensate for the reduced model capacity and even substantially improve upon the strong nnU-Net baseline. This is represented also in most individual class metrics, in both the DSC and NSD. Only the right adrenal gland shows performance which is markedly worse than the baseline.

This performance is also reflected in the held-out test set, where the proposed noisy nnU-Net student achieved 87.4 DSC and 91.68 NSD, which is in line with the validation results, if not slightly better. For this reason, we believe the validation results are a reliable indicator of general performance gains with the proposed method.

**Table 7.** Quantitative results on the validation dataset in terms of mean DSC/NSD per class and overall. The best results per metric are highlighted in bold.

| Class | nnU-Net | | low-footprint nnU-Net | | noisy nnU-Net student | |
|---|---|---|---|---|---|---|
| | DSC | NSD | DSC | NSD | DSC | NSD |
| Liver | 96.73 | 94.33 | 96.5 | 94.43 | **97.58** | **97.43** |
| Right Kidney | 85.2 | 83.59 | 80.44 | 77.3 | **86.61** | **85.88** |
| Spleen | 93.87 | 92.25 | 92.45 | 91.16 | **95.75** | **95.32** |
| Pancreas | 82.11 | 91.04 | 83.86 | 92.4 | **84.05** | **93.4** |
| Aorta | **97.13** | 98.52 | 96.96 | 98.25 | 96.58 | **98.8** |
| Inferior Vena Cava | 87.64 | 87.25 | 87.22 | 87.02 | **88.17** | **88.36** |
| Right Adrenal Gland | **85.43** | **94.95** | 80.27 | 89.73 | 80.73 | 90.94 |
| Left Adrenal Gland | **88.23** | 96.96 | 82.7 | 91.71 | 88.13 | **97.57** |
| Gallbladder | 57.19 | 56.91 | 62.95 | 62.0 | **67.18** | **66.31** |
| Esophagus | 86.01 | 92.72 | 83.92 | 90.3 | **88.56** | **96.16** |
| Stomach | 87.82 | 88.41 | 80.82 | 84.68 | **88.4** | **90.95** |
| Duodenum | 71.94 | 84.83 | 74.07 | 86.74 | **77.75** | **89.7** |
| Left Kidney | 86.2 | 85.73 | 82.51 | 79.83 | **91.13** | **91.01** |
| Mean | 85.04 | 88.27 | 83.44 | 86.58 | **86.97** | **90.91** |

### 4.3   Qualitative results

We present segmentation results on an easy, as well as a hard sample in Figure 2. In the top row we can attest the proposed method a clear improvement over the low-footprint model without using the unlabeled data via noisy student training, while showing similar shortcomings as the original nnU-Net, as can be seen e.g. from the class confusion in the lower left side of the shown segmentation maps. As the example in the bottom row of Figure 2 shows however, the model is still susceptible to confusing e.g. the left and right kidney. Such failure cases could potentially be decreased by more involved postprocessing and ensembling of multiple models, which in turn comes at the cost of slower inference speeds.

## 5   Conclusion

We conclude that the proposed method of self-training with a noisy student can lead to performance improvements even over strong baselines such as the nnU-

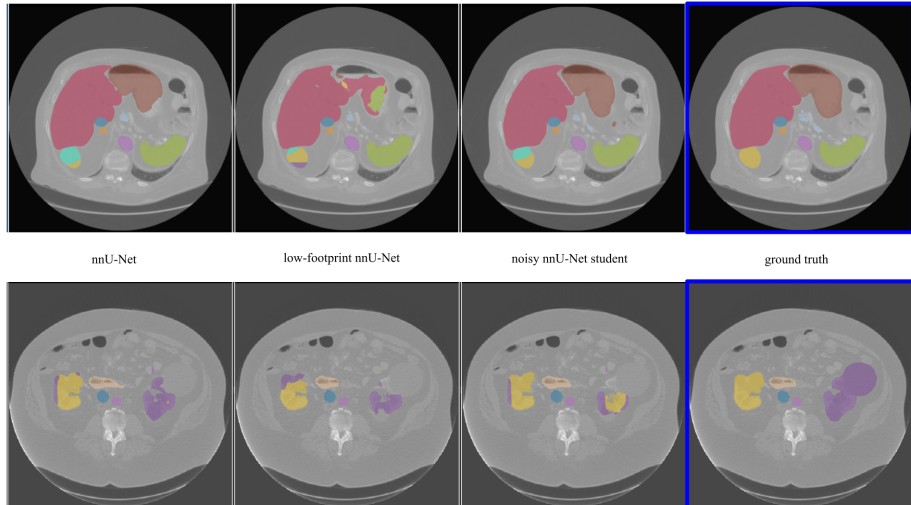

**Fig. 2.** Qualitative results on a rather easy (case 2, top) and a rather hard (case 23, bottom) example from the validation set. The first column shows the prediction by nnU-Net, the second column the prediction by the low-footprint model, the third column the proposed method's prediction and the fourth column shows the ground truth labels.

Net [5]. While we restricted ourselves to a low memory footprint architecture, we note that performance improvements might be more pronounced when employing larger teacher and student models or longer schedules with stronger data augmentation. We also note that future work could incorporate segmentation confidence estimates in order to filter for high-confidence pseudo labels.

**Acknowledgements** The authors of this paper declare that the segmentation method they implemented for participation in the FLARE 2022 challenge has not used any pre-trained models nor additional datasets other than those provided by the organizers.

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
