# OpenReview forum: "A Noisy nnU-Net Student for Semi-supervised Abdominal Organ Segmentation"
_MICCAI.org/2022/Challenge/FLARE_

### Official Review · Reviewer_WnPV · 2022-09-17
**Points are clear generally speaking, just need some minor changes**

**Rating:** 7
**Confidence:** 4

**Review:**

The paper showed their points clearly, and writing in such manner that it stick with their list of contributions. That's being said, the structure of paper is coherence. However, I have some advice about content arrangement.

Abstract: Author articulate the challenges and the method abundantly. But, about the results, I feel it is sketchy. Maybe need to include quantitative results, like model achieve # DSC, using # GPU memory, inference speed reduce from # time to # time by factor of two.

In the method, "We propose a method based on the nnU-Net framework [5]". And in 2.2 Proposed Method, "The proposed noisy nnU-Net student is inspired by knowledge distillation [4]". I guessing author is trying to say that we propose noisy nnU-Net student inspired by KD[4]
, nnU-Net framework[5]. If it articulate in this manner, it's more coherence and easy to read.

I did not spot any grammar or other issues in this paper further

---

> ### Author Response · Authors · 2022-10-13
> **Thanks for the valuable feedback!**
>
> We thank reviewer “WnPV” for the valuable feedback! Based on the reviewer’s comments, we adapted the abstract, as well as the mentioned part in the method section.

---

### Official Review · Reviewer_CttY · 2022-09-18
**A nice try to combine nnU-Net with nosiy student**

**Rating:** 9
**Confidence:** 3

**Review:**


**Summary**
This paper combine nnU-Net with noisy student to use unlabeled data and design a low-footprint nnU-Net to achieve efficient inference.

**Pros**
* Well-written paper with clear method description and complete experimental results.
* Simple but effective method to make low footprint nnU-Net student yields significant improvement over default nnU-Net but consume less GPU memory and inference 10× faster.

**Problems**
* The experiments are conducted on only 20 held-out cases, it is better to report the results on validation leaderboard (50 cases) to compare with other participants methods directly.
* For better visualization of Fig2, it is better to adjust the `level` and `window` to be consistent and all images should have the same image size.

---

> ### Author Response · Authors · 2022-10-13
> **Thanks for the very kind feedback!**
>
> We thank reviewer “CttY” for the very kind feedback! We adjusted Fig. 2 to better visualize the images in a consistent way. Concerning the ablations on 20 held-out cases, we understand the wish for a larger test set and hence more robust conclusions. However, as the challenge is closed, we cannot submit all ablations on the validation leaderboard. In case validation submissions are re-opened, we can supply these results later on.

---

### Official Review · Reviewer_ha3X · 2022-09-19
**A efficient and effective method**

**Rating:** 8
**Confidence:** 3

**Review:**

This paper exploits an ensemble of teacher networks to provide pseudo-labels for training a noisy nnU-Net student. To reduce computational costs, this paper further uses a small architecture and resampling scheme. Experimental results show that the proposed method can effectively save computational costs and improve segmentation performance.




Pros:

+ This paper adapts nnU-Net to semi-supervised learning setting by using self-training with an ensemble of teacher models and a noisy student model of
the same network architecture.

+ This paper achieves good performance and effectively reduces computational costs.

+ The ablation studies shows the effectiveness of design choices used in the proposed methods.

Cons:

- The introduction section presents several challenges such as shape variations, abnormalities, and distribution shifts. However, these challenges are not the focus of this paper as the method section does not clearly clarify how these challenges are addressed. A better way is to focus on the unlabeled data and the efficiency in FLARE 22.

- In the Introduction, "With some small architecture and preprocessing adaptations". But the pre-processing in Sec. 2.1 is too short to present the preprocessing adaptations. So I would suggest including some more details, e.g., how to do percentile clipping.

- In Tab. 3, please explain a little bit about what p_{sample} and P_{channel} are.

---

> ### Author Response · Authors · 2022-10-13
> **Thanks for the kind feedback!**
>
> We thank reviewer “ha3X” for the kind feedback! Based on the feedback, we rewrote the introduction section and added additional information in table 4 w.r.t. the data augmentation probabilities, as well as in Sec. 2.1, concerning global percentile clipping.

---

### Official Review · Reviewer_Dxnd · 2022-09-20
**Review for Noisy nnU-Net**

**Rating:** 7
**Confidence:** 4

**Review:**

The authors apply noisy student model to achieve efficient abdominal organ segmentation. In addition to the model construction, they also made a series of data pre-processing and post-processing to lower the computational cost. The reported results show that the proposed method achieves better performance and lower cost than the original nn-UNet.

Advantages:
1. The paper is written fluently and is easy to understand.
2. Authors optimize the entire working flow such that the proposed method has faster inference speed and lower computational cost.
3. The quantitative comparison is very detailed.

Disadvantage:
1. The novelty of this method is relatively weak. It seems the authors just use the noisy student model without proposing new ideas or methods.
2. Authors mentioned that they ensembled 5 teacher models but did not mention the ensembling method and the ablation study on the teacher model is not clear. For example, why use 5 teachers? What is the performance of the teachers?

---

> ### Author Response · Authors · 2022-10-13
> **Thanks for the constructive feedback!**
>
> We thank reviewer “Dxnd” for the constructive feedback! We added more details w.r.t. the teacher model’s performance and the ensembling method used in the revised paper. Concerning the novelty, our aim with this challenge was rather establishing which performance can be achieved in this setting, while relying on straight-forward methodology with minimal assumptions regarding the task. Still, combining noisy student training with nnU-Net, to the best of our knowledge, has not been officially tested before. With this in mind, the novelty in our proposed solution lies more in the inference optimization, which allows speeding up inference by an order of magnitude on CT scans comparable to the ones encountered in this challenge.

---

### Meta-Review · Program_Chairs · 2022-09-28

**Recommendation:** Minor Revision
**Confidence:** 5

**Metareview:**

Nice paper. Please address the reviewers' comments in the revised manuscript